# Ultra-Short Lifetime of Intersubband Electrons in Resonance to GaN-Based LO-Phonons at 92 meV

**Daniel Hofstetter** [1],* , **Hans Beck** [2] and **David P. Bour** [3]

1   Independent Researcher, Chemin du Château 5, 2068 Hauterive, Switzerland
2   Independent Researcher, Rue des Peupliers 6, 2014 Bôle, Switzerland; hans.beck39@bluewin.ch
3   Google LLC., 1250 Reliance Way, Fremont, CA 94539, USA; davidpbour@gmail.com
*   Correspondence: dani_hofstetter@outlook.com; Tel.: +41-32-753-78-25

**Abstract:** In this study, we report on the ultra-short lifetime of excited intersubband electrons in a 38 Å wide AlGaN/GaN-based quantum well. The rapid decay of these charge carriers occurs due to a resonance between the relevant intersubband transition energy and the size of the GaN-based LO-phonon at 92 meV. Based on the experimentally observed Lorentz-shaped intersubband emission peak with a spectral width of roughly 6 meV ($48 \text{ cm}^{-1}$) respecting the Fourier transform limit, a very short lifetime, namely 111 fs, could be calculated. By comparing this lifetime to the existing literature data, our value confirms the potential high-speed capability of III-nitride-based optoelectronics.

**Keywords:** GaN; quantum cascade structure; intersubband transition; resonance; optical emission; LO-phonon frequency; large FWHM; short lifetime

## 1. Introduction

Although the wide-bandgap semiconductor GaN was synthesized and characterized for the first time as long ago as 1931 [1], improving its crystalline quality was initially not at all straight-forward [2,3]. Since 1990, when researchers at Nichia, Ltd. started to make rapid advances regarding the epitaxial growth of III-nitride compounds [4], this wide-bandgap semiconductor has become subject to intense research again. Besides several fundamental properties, the main emphasis of these efforts quickly shifted toward the demonstration of powerful high electron-mobility transistors (HEMTs) [5] and efficient light emitters, such as white light-emitting diodes (LEDs) [6] or violet–blue laser diodes [7]. As an interesting alternative to these mainstream topics, optical absorption using AlGaN/GaN-based intersubband structures was demonstrated in 2000 [8,9]. Mainly because of technological issues (substrate, strain, doping, etc.), further progress was slow. Nonetheless, the first near-infrared photo-detectors using GaN-based intersubband transitions were presented in 2003 [10]. In the following 20 years, our group regularly published papers in the field of III-nitride semiconductors. In most cases, topics related to intersubband optoelectronics were investigated [11,12].

In the present work, we take advantage of an optical measurement technique to obtain a precise value for the lifetime of excited electrons in an AlGaN/GaN-based quantum cascade (QC) structure. Its vertical electronic transition takes place in a 38 Å wide GaN quantum well that constitutes the 'active region'. Under application of a suitably sized electric field, the principal electronic transition in the structure's main quantum well can—due to a pronounced quantum-confined Stark effect—be tuned into resonance with the GaN-based LO-phonon at 92 meV [13]. We then observe a 6 meV wide emission peak, whose Fourier transform limited pulse length—in agreement with Heisenberg's uncertainty principle in Ref. [14]—is extremely short; i.e., 111 fs. This configuration resembles the lower energy level of a QC laser, where a one- or two-phonon-resonance also results in a very short lifetime. The size of this fundamental material parameter is compared to measurement values obtained using alternate methods and semiconductors. Among these

numbers, there is, for instance, an LO-phonon linewidth value published in 1999 by A. Link et al. in the *Journal of Applied Physics* [15]. Using Raman spectroscopy, the authors of this study obtained linewidths of 4 cm$^{-1}$ at 77 K and 4.5 cm$^{-1}$ at 200 K. At the same temperature values, the absolute peak position shifted from 737 cm$^{-1}$ (77 K) to 735 cm$^{-1}$ (200 K). Several other authors investigated electron scattering rates and lifetimes using LO-phonon resonance in various material systems [16,17]. The observed values of 0.3 ps for GaAs/AlGaAs [18], 0.35 ps for InGaAs/InAlAs [19], 0.38 ps for GaInAs/AlAsSb [20], and 0.45 ps for the ZnO/MgO [21] material system are all relatively close together and, thus, indicate a relatively weak dependence on the bandgap of the involved semiconductors. In this article, we will present an optoelectronic method that we used to determine the lifetime of excited electrons in an AlGaN/GaN quantum well-based intersubband transition. As could be expected from the involved high-bandgap materials, their resonance with the GaN LO-phonon results in a particularly short lifetime. In agreement with our own measurements [22], the presented results clearly and unambiguously confirm the ultra-high-speed potential of GaN-based optoelectronic devices.

## 2. Device Fabrication and Experimental Methods

The investigated structure was fabricated using metal-organic vapor phase epitaxy (MOVPE) on a c-face sapphire substrate. It started with a 5 μm thick and n-doped GaN buffer layer (Si, $5 \times 10^{17}$ cm$^{-3}$), on top of which a 5-period superlattice (SL) was grown. The final layer of the structure was a 200 nm thick top contact, which consisted of highly n-doped GaN (Si, $1 \times 10^{18}$ cm$^{-3}$). The 180.5 nm thick active region consisted of a SL and was composed of GaN quantum wells (QWs) and AlGaN barriers, having an Al-content of 24.5%. Similar to a QC laser, its injector (roughly 40% of the total layer thickness) was n-doped (Si, $5 \times 10^{17}$ cm$^{-3}$). Since the exact layer sequence has already been published, we only refer to some of the corresponding papers [23–25].

In Figure 1, we show two overview emission spectra (B1767a2 and B1976a1) measured using a Fourier transform infrared (FTIR) spectrometer. We used 2.5 μs long current pulses at a 100 kHz pulse repetition rate (25% duty cycle) to pump these devices. Due to a relatively low doping level (typical resistance >10 MΩ at 77 K for a square-shaped mesa with 200 μm side-length), an elevated bias voltage of 18.5 V was necessary to drive a moderate injection current of 156 mA through these devices. For this reason, severe device heating occurred. This temperature increase mainly manifested itself via the generation of a substantial number of phonons. Among these different types of lattice vibrations, there was also a large number of LO-phonons, which had an energy of 92 meV at the Brillouin zone's centre. At the same time, the necessary high electric field resulted in a marked quantum-confined Stark effect [11]. At larger electric fields, the energy difference ($\Delta E = E_1 - E_0$) between the two lowest energy levels in the 38 Å-wide main QW gradually diminished and, finally, became exactly resonant to the GaN LO-phonon.

Figure 2 schematically shows this resonance configuration. It is obvious that the quantized lattice vibrations were directly generated through the heat produced by the high operating voltages (18.5 V/6.3 V for B1767a2/B1976a1, respectively), injection currents (156 mA/400 mA), and duty cycles (25%/50%). At their decay, these LO-phonons resonantly transmitted their energy onto intersubband electrons. The electrons were, in turn, lifted from the ground level ($E_0$) to the first excited energy level ($E_1$) of the main QW. At the end of this quantum mechanical series of events, the excited electrons decayed from the upper ($E_1$) to the lower ($E_0$) energy level of the main GaN QW, thereby emitting photons with an energy of 92 meV.

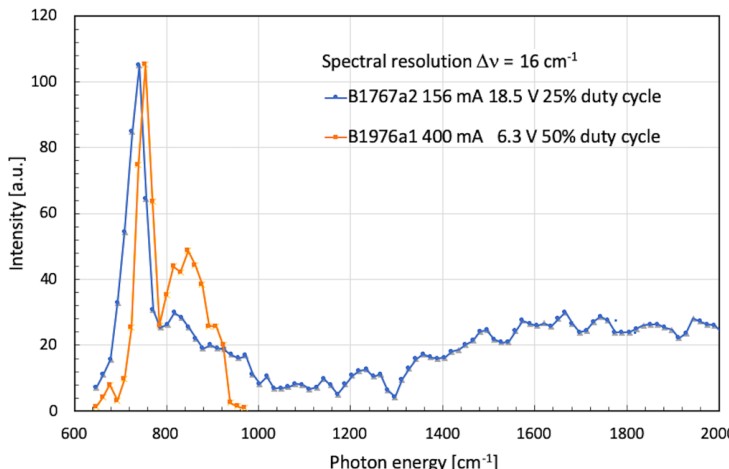

**Figure 1.** Intensity of optical emission as a function of photon energy for devices B1767a2 and B1976a1 under electrical injection. In these two overview spectra, the peaks at the frequency of the GaN-based LO-phonon at $741/755$ cm$^{-1}$ are clearly visible.

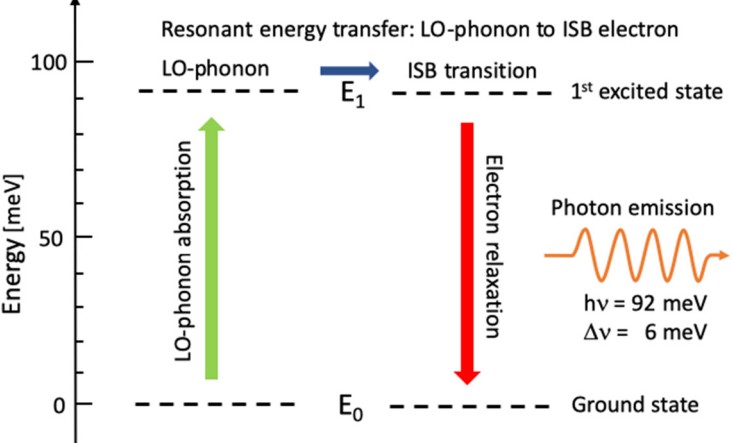

**Figure 2.** Schematic energy diagram showing the resonance between the GaN-based LO-phonon and the intersubband transition involving the two lowest QW states ($E_0$ and $E_1$). For samples B1767a2 and B1976a1, this resonance condition was exactly fulfilled at applied voltages of 18.5 and 6.3 V.

Using the injection conditions described above, we measured the optical emission spectrum of the main well's fundamental transition ($E_1 \rightarrow E_0$, $\Delta E = 92$ meV $= 741$ cm$^{-1}$) and observed a full width at half maximum (FWHM) of $\Delta\nu = 6$ meV (48 cm$^{-1}$). During these experiments, the samples were mounted onto the cold finger of a liquid nitrogen-cooled flow cryostat ('Janis ST-100-MOD') and stabilized at a temperature of 77 K. In close proximity to the active mesa of the sample, an overhanging ($45°$ inclined) outcoupling facet was polished. Since the luminescence signal was not very strong, an advanced measurement technique had to be used. The injection current was delivered using an 'Agilent 8114A' high-power pulse generator. Typical pulse lengths of 2.5 µs or 5 µs at a repetition frequency of 100 kHz and voltages of 20 V and 6.3 V (for samples B1767a2 and B1976a1, respectively) were used. The luminescence measurements were carried out using a Fourier transform infrared spectrometer (FTIR, Nicolet Magna 800) and based on 45-min-long step-scans using lock-in amplification (Stanford Research Systems SR 865A). During such step-scans, the movable mirror of the FTIR's Michelson interferometer was held at a fixed position for 2 s, before stepping to the next (fixed) position. Due to this rather time-consuming measurement procedure, the mutual point-to-point separation was limited to a relatively coarse value of 16 cm$^{-1}$ (precise number: 15.4277 cm$^{-1}$), instead

of the technically possible value of 0.125 cm$^{-1}$ (precise number: 0.1205 cm$^{-1}$). We then carefully examined this FTIR-measurement around the GaN-based LO-phonon frequency of 741 cm$^{-1}$.

For this purpose, as shown in Figure 3, we used the corresponding peak, including every single measurement point. Since the optical emission at the LO-phonon frequency had a FWHM of almost four (i.e., not the theoretically minimal single) horizontal point-to-point distances, it was clear that even a considerably better spectral resolution could *not* have resulted in a substantially narrower linewidth. The optical resolution was, thus, set at the necessary minimum, rather than at the possible minimum. However, it could not be discounted that effects like thermal chirping due to pulse-related heating of the device contributed to an additional spectral broadening. To estimate such parasitic effects more accurately, the frequency behavior of the GaN-based LO-phonon had to be examined carefully. According to Figure 3b in the paper of Link et al. [15], one expects—between 77 K and an assumed maximum active region temperature of 200 K—a very minor broadening of 2 cm$^{-1}$ (i.e., from 4 cm$^{-1}$ to 6 cm$^{-1}$). At the same time, we see in Figure 3a of [15] a small wavelength shift from 737 cm$^{-1}$ (77 K) to 736 cm$^{-1}$ (200 K). Finally, Viswanath et al. found that the exciton–phonon scattering effect also leads to some broadening of this optical emission peak. In Ref. [26], a value of 4 cm$^{-1}$ was cited. Taken together and assuming simple linearity, these effects account for a small broadening on the order of 7 cm$^{-1}$ of the peak near the LO-phonon frequency. Based on these arguments, the resulting line width in our experiment was slightly reduced, but it still remained as high as 48 cm$^{-1}$ (6 meV).

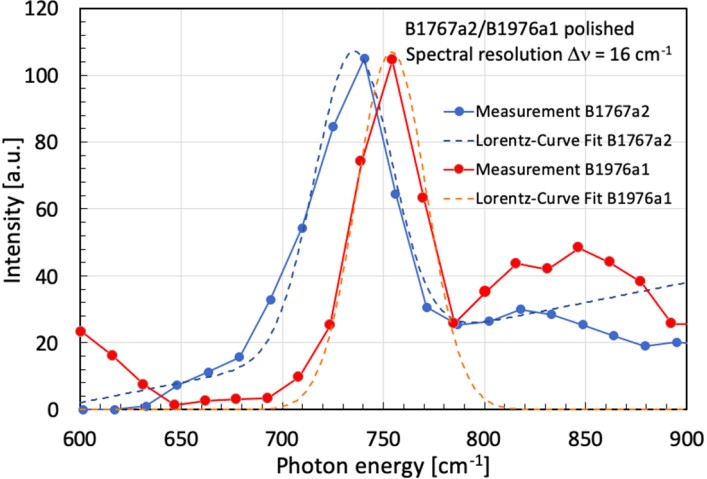

**Figure 3.** Close-up view of the optical spectra of samples B1767a2 and B1976a1. The final photon emission close to the LO-phonon frequency can be fitted using a Lorentz-curve peaking at 735 cm$^{-1}$ and 754 cm$^{-1}$ and with respective FWHMs of 50 and 40 cm$^{-1}$. The fit of the B1767a2 sample includes a linear slope, and it runs from 650 to 850 cm$^{-1}$. In contrast, the one to B1976a1 runs between 650 and 780 cm$^{-1}$.

## 3. Results

Our measurement of the relatively broad linewidth is quite interesting and has some important implications. It means that a more fundamental effect is the origin of the observed linewidth broadening of the LO-phonon. This observation, however, leads us to look for possible causes of this rather large peak width of 6 meV. After all, each constituting atom of this GaN crystal has exactly the same mass and size. In addition, these atoms are arranged in a highly periodic lattice of low crystalline defect density, and the interactions between them are absolutely identical. All of these points are prerequisites for finding a very high quality factor of the corresponding lattice vibrations. Therefore, in the absence of any other good explanation, the large FWHM of this peak around the GaN-based LO-phonon frequency must have other causes. In Figure 4, this resonance between the intersubband transition and the LO-phonon is illustrated.

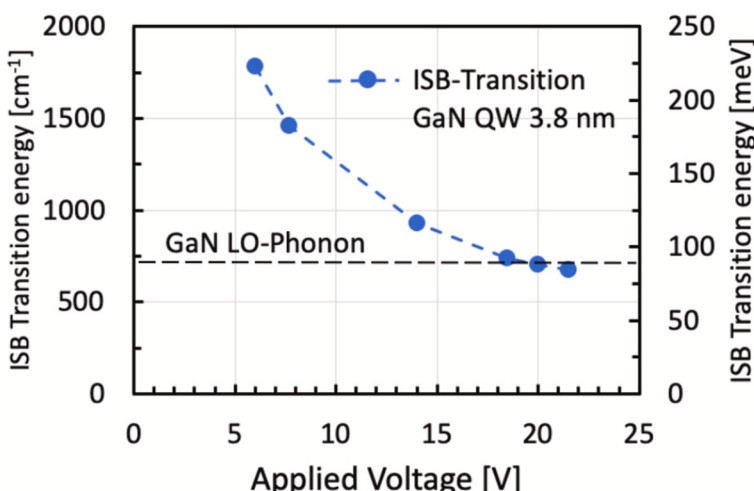

**Figure 4.** LO-Phonon in resonance with GaN-based intersubband transition in the 38 Å wide main QW of the presented QC structure. This figure is a re-publication of Figure 3 in [25].

Since it is well known that excited electrons in QW-based intersubband transitions have short lifetimes, and since the ISB transition in the present situation is also resonant to the LO-phonon, an ultra-short lifetime can actually be expected. Based on the measured (Lorentzian) peak width of $\Delta\nu_{\text{Lorentz}} = 48$ cm$^{-1} = 1.439 \times 10^{12}$ s$^{-1}$, we are able to calculate the corresponding electron lifetime $\Delta\tau_{\text{life}}$ using the following Equation (1), which can also be found as Equation (10) in the paper of Cuscó et al. [21]:

$$\Delta E \cdot \Delta\tau_{\text{life}} = \hbar \tag{1}$$

If we expand $\Delta E$, solve for $\Delta\tau_{\text{life}}$, and reduce the resulting fraction by '$h$', we obtain the following simplified expression:

$$\Delta\tau_{\text{life}} = \frac{\hbar}{\Delta E} = \frac{h/(2\pi)}{h\Delta\nu_{\text{Lorentz}}} = \frac{1/(2\pi)}{\Delta\nu_{\text{Lorentz}}} \tag{2}$$

Using the above value of $\Delta\nu_{\text{Lorentz}} = 1.439 \times 10^{12}$ s$^{-1}$, we calculate a lifetime of $\Delta\tau_{\text{life}} = 111$ fs. Given some experimental uncertainty, this value aligns well with the corresponding (available) lifetimes determined by other researchers. Compared to the LO-phonon lifetimes in other material systems, we notice that the value reported in this work is considerably smaller. This fact is clearly visible in Table 1, in which several experimentally observed intersubband electron lifetimes—typically measured at LO-phonon resonance—are listed. As an illustrative example, a value of 0.35 ps was reported for the InGaAs/InAlAs material system by Sirtori et al. in 1994. Two years later, Sirtori et al. published an electron lifetime for AlGaAs/GaAs. A slightly shorter value of 0.3 ps was measured. This difference is a clear indication of the larger covalent bonding of GaAs compared to InGaAs. Obviously, the larger hardness of GaAs with respect to InGaAs is also reflected in the larger bandgap and the higher LO-phonon energy of GaAs. In particular, we find, for GaAs/AlGaAs (GaAs bandgap: 1.43 eV), a value of 36 meV, while for InGaAs/InAlAs (InGaAs bandgap: 1.02 eV), a value of 32 meV was obtained. In the meantime, LO-phonons in a couple of other semiconductors have been investigated. Among them, the following materials are listed in Table 1.

**Table 1.** List of common semiconductor materials, as well as their bandgaps, LO-phonon energies, and—where available—excited electron lifetimes at LO-phonon resonance. The last column contains references to papers that discuss their most important physical properties. The main reason for the values being missing is that these semiconductors are not typical QW materials.

| Material | Bandgap | LO-Phonon | Excited Electron Lifetime | Reference |
|:---:|:---:|:---:|:---:|:---:|
| C | 5.47 eV | 167 meV | - | [27] |
| GaN | 3.41 eV | 92 meV | 0.17 ps | [9] |
| ZnO | 3.27 eV | 72 meV | - | [21] |
| GaP | 2.32 eV | 50 meV | - | [28] |
| AlAs | 2.12 eV | 50 meV | - | [28] |
| CdSe | 1.74 eV | 26 meV | - | [29] |
| GaAs | 1.43 eV | 36 meV | 0.3 ps | [30] |
| InP | 1.42 eV | 43 meV | - | [28] |
| InGaAs | 1.02 eV | 32 meV | 0.35 ps | [19] |
| InAs | 0.43 eV | 30 meV | - | [28] |
| PbTe | 0.32 eV | 13 meV | - | [31] |
| InSb | 0.17 eV | 24 meV | 0.38 ps | [28] |

Obviously, the lifetime values listed in Table 1 confirm our above findings that hard materials have high bandgaps and large LO-phonon energies, while soft materials have opposite traits.

## 4. Conclusions

In the presented experiments, we have shown a precise and indirect measurement of the LO-phonon-governed excited electron lifetime in AlGaN/GaN-based semiconductors. We obtained a lifetime of $\tau_{\text{life}} = 111$ fs, which is roughly a factor of three smaller than that seen in other semiconductors [32–35]. As a matter of fact, those semiconductors with high bandgap energies are typically among the hardest known materials, such as diamond or GaN. Likewise, we also observed that such strongly bonded semiconductors exhibit large LO-phonon energies. As our short listing of the relevant parameters in Table 1 shows, the high covalent bonding strength of these materials results not only in high bandgaps, but also in very short lifetimes of excited QW electrons [36]. This result is especially true in the case of a resonance between LO-phonon and intersubband transition. Therefore, the presented measurements of an ultra-short electron lifetime in GaN QWs align perfectly well with the results cited above. Even more importantly, they pave the way for innovative high-speed solutions using intersubband devices based on AlN/GaN superlattices.

**Author Contributions:** Conceptualization, D.H. and H.B.; methodology, D.H.; software, D.H.; Writing—Original draft, D.H.; Writing—Review and editing, D.H., H.B. and D.P.B. All authors have read and agreed to the published version of the manuscript.

**Funding:** This research received no external funding.

**Institutional Review Board Statement:** Not applicable.

**Informed Consent Statement:** Not applicable.

**Data Availability Statement:** The original data are available from D.H., H.B. and D.P.B. on request.

**Acknowledgments:** The authors would like to thank the former Institute of Physics at the University of Neuchâtel for making available their experimental equipment, express gratitude for the financial support provided by the Professorship Program of the Swiss National Science Foundation, and acknowledge the valuable discussions with Klaus Reimann of the Max-Born Institute in Berlin.

**Conflicts of Interest:** The authors declare no conflict of interest.

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
