# Peer review of "Ultra-Short Lifetime of Intersubband Electrons in Resonance to GaN-Based LO-Phonons at 92 meV"

_photonics, doi:10.3390/photonics10080909_

Round 1

Reviewer 1 Report

In this study, the researchers investigated the emission spectrum of sub-band transitions in GaN-based quantum wells, specifically focusing on the transition energy resonating with LO-phonons. It was observed that the spectral FWHM of the emission is notably broad, approximately 6 meV, indicative of an exceptionally short lifetime, estimated at 107 fs. While these findings are interesting, it is recommended that major revisions be made to the manuscript prior to publication.

1)      The study should include controlled experiments. The authors are recommended to measure the emission spectrum when the electronic transition does not resonate with the LO phonons. According to their explanation, such a non-resonant condition should result in a narrower emission peak, which will also serve to confine the instrumental broadening, in addition to the discussed parasitic broadening.

2)      The discourse spanning lines 166 – 182 lacks clarity in its focal point. The authors reference previous studies concerning electron lifetime, LO-phonon lifetime, LO-phonon energy, and atomic bonding strength. However, the connection between these properties and the findings of the present study is not clearly elucidated. Moreover, the reference to a 'universal scaling law behind this behavior' (line 189) requires additional clarification. Introduction of relevant formulas or expanded discussion would be helpful.

3)      The manuscript appears to be missing a clear conclusion. What are the implications and insights derived from this study?

Additionally, there are some minor issues to address:

1)      More detailed explanation of the measurement method is necessary. Specifically, the parameters of the lock-in amplification should be provided.

2)      Why is the x axis of figure 2 labeled Time?

3)      Usage of exclamation marks in the text should be avoided to maintain a formal and academic tone (line 123; line 150).

4)  The authors are advised to use declarative sentences rather than interrogative ones. (line 144; line 151).

Please refer to the minor points in the comments.

Author Response

See attached file 'Reviewer #1 - MDPI - photonics - 2484600 - response.docx' below.

Reviewer 2 Report

This manuscript presents findings on the ultra-short lifetime of excited electrons in a 38 Å wide AlGaN/GaN-based quantum well. The authors propose that this short lifetime arises from the resonance between the GaN-based LO-phonon at 92 meV and the inter-subbandtransition. However, the manuscript lacks organization and requires a more comprehensive and thorough analysis for stronger conclusions. Below, I provide comments and raise questions regarding this article. Overall, considering the manuscript's deficiency in terms of originality and persuasiveness, I do not recommend it to be published in photonics.

1. The introduction does not sufficiently highlight the significance of the research conducted by the authors. It is essential for the authors to clarify the motivation behind obtaining the lifetime value in the materials. Additionally, the most recent relevant works should be mentioned in the introduction.

2. The manuscript should provide details regarding the device structure, measurement setup, and experimental procedures. 

3. The experimental investigation predominantly concentrated on the sample B1767a2. However, to mitigate the potential influence of sample instability, it is imperative to conduct tests using several additional samples. This approach would enhance the reliability and robustness of the findings.

4. The derivation process and calculation used to determine the electron lifetime lack persuasiveness, which is crucial for understanding your paper. To address this concern, it is recommended that the authors provide physical derivations or visual representations that clarify the relationship between the resonance of the inter-subband transition and the LO-phonon and the resulting reduction in electron lifetime.

5. Overall. the conclusion and analysis presented in the manuscript primarily rely on the optical spectrum depicted in Figure 1, which is insufficient to provide a comprehensive understanding. To ensure the robustness of their findings, the authors should augment their analysis by including additional data that supports their conclusion.

Author Response

Please find my answers in the file 'Reviewer #2 - MDPI - photonics - 2484600 - response.docx' attached below

Reviewer 3 Report

Hofstetter et al observed an ultrashort lifetime, about 10fs, of inter-subband electrons in an AlGaN/GaN quantum well. These excited electrons are produced via resonance coupling with the GaN based LO phonons when the quantum well is pumped. This result is interesting because it suggests the possibility of high-speed electronics using III-nitride quantum well designs. Some comments are listed below:

1/ line 34, since the measurement technique is not the focus of the paper, it’s best to explain what technique is used here than keeping it a mystery. 

2/ please clarify the name “device B1767a2” – how many samples are tested to obtain the results? What’s the expected variation among samples ?

3/ line 94, why is “obvious” phonons are generated by voltage, current and duty cycle? What energy phonons can one expect if the voltage and duty cycles are varied?

4/ line 170 to the end of result section: this part is already largely mentioned in the introduction. Please rewrite not to repeat this information in the paper.

Good. 

Author Response

Please find my answers in the file 'Reviewer #3 - MDPI - photonics - 2484600 - response.docx' attached below.

Round 2

Reviewer 1 Report

The quality of the manuscript has been improved. I suggest its publication.